# Strength and Cyclic Properties of Additive vs. Conventionally Produced Material AlSi_10_Mg

**DOI:** 10.3390/ma16072598

**Published:** 2023-03-24

**Authors:** Vladimír Chmelko, Miroslav Šulko, Jaroslava Škriniarová, Matúš Margetin, Marek Gašparík, Tomáš Koščo, Marián Semeš

**Affiliations:** 1Institute of Applied Mechanics and Mechatronics, Slovak University of Technology in Bratislava, Námestie Slobody 17, 81231 Bratislava, Slovakia; 2Institute of Informatics, Slovak Academy of Sciences, Dúbravská Cesta 9, 84507 Bratislava, Slovakia

**Keywords:** aluminium alloy AlSi_10_Mg, additive manufacturing, conventional casting, cyclic properties

## Abstract

Additive metals are practically identical in strength to the properties of conventionally produced materials. This article experimentally analyses strength properties and fatigue properties in the tensile–pressure mode for two different directions of 3D printing of AlSi_10_Mg material. The resulting fatigue parameters of the Basquin curve are confronted with a conventionally produced alloy of the same composition. The microstructure analysis explains the different fatigue properties obtained by these two material production technologies. Phenomena such as strength enhancement in additive manufacturing material, anisotropy of cyclic properties, and cyclic hardening are discussed. The limits of current additive manufacturing are clarified, and the future direction of research in this field is outlined.

## 1. Introduction

In the last decade, the production of components by the gradual addition of material volume, referred to as additive manufacturing (AM), has been increasingly applied in the production of metal components [1,2,3]. Powder sintering or jetting opens up new possibilities for component shapes of external complexity or internal granularity in a wide range of applications [4,5]. For components that require a computational assessment of their strength (static integrity or fatigue strength), it is necessary to know the mechanical properties of the material produced by additive manufacturing. For engineering simulations and calculations, it is necessary to know at least Young’s modulus E and Poisson’s ratio for defining the parameters of the linear model. Minimum S-N curves are needed to assess the fatigue properties and fatigue strength of the additive manufacturing material [6,7]. Comparing conventional welding technology with AM technology, the welding process causes the formation of liquid metal at the point of maximum electrical energy input and thermal influence on the adjacent layers of the materials being joined. These metallurgical processes are now relatively well studied, resulting in virtually identical mechanical properties (E) compared to the base material. The microstructure affected by the thermal processes has significantly reduced cyclic properties, which is also reasonably well documented experimentally [8,9,10]. In contrast, additive manufacturing, e.g., in the form of laser beam sintering of powders, produces complex multiphase ratios at the sintering site where the solid phase, the gas phase, and the molten phase coexist. This condition, together with the high cooling gradients and reflow of the material layers as the beam passes through the new added layer, is not yet clearly described physically [9,11,12]. Therefore, the strength of the bonds formed may not automatically be the same when comparing AM production and conventional metallurgy.

Aluminium alloys are among the materials often used in practice for casting complex-shaped parts. Several papers have been published on the selected mechanical properties of aluminium alloys produced by additive manufacturing [13,14,15]. This paper focuses on the AlSi_10_Mg alloy. It presents and summarizes the basic tensile curve parameters of AlSi_10_Mg materials obtained by conventional metallurgy and additive manufacturing as well as their cyclic properties in the form of S-N curves are compared. The relation of these properties to the built direction of the layers is also analysed. Thus, this work provides comprehensive information for both strength and fatigue assessment of additively manufactured components made from AlSi_10_Mg alloy in the as-built condition.

## 2. Materials and Methods

AlSi_10_Mg alloy is practically the most widely used aluminium alloy for additive manufacturing by SLM (selective laser melting) [15,16,17]. The materials of this type (silumines) are used to produce, for example, casings for combustion engines and transmissions, wheel discs, and components for the aerospace industry. The material is characterized by good foundry properties, weldability, and good thermal conductivity. The intermetallic Mg_2_Si particles subject the base matrix to precipitation strengthening without deteriorating other properties.

In this study, two AlSi_10_Mg materials, whose chemical compositions are virtually identical (Table 1) but were produced by different technologies, are compared. The additively manufactured samples were printed on a Concept Laser Xline 2000R (laser head power is 1000 W, and sintered layer thickness is 0.05 mm). The material produced by conventional metallurgy was obtained directly from the smelter of SLOVALCO, j.s.c.

Metallographic analysis of the microstructure of the identical chemical composition of the material produced by AM technology and conventional metallurgy reveals differences [18,19,20]. The microstructures of both materials are shown in Figure 1 with a detail in the upper left hand. The microstructure of the conventionally cast AlSi_10_Mg material (Figure 1) is formed by a base matrix of a solid solution of Si in Al. A eutectic formed by the solid solution of Al and Si crystallized along the primary grain boundaries. The microstructure contains intermetallic Mg_2_Si particles with a strengthening effect. The microstructure of the additively manufactured material in Figure 1b is formed by a supersaturated homogeneous solid solution without eutectic of Si in Al with visible melting pools due to the high cooling rate. This first visual comparison of the materials leads to the expectation of differences in their mechanical properties.

## 3. Results

### 3.1. Analysis of AlSi_10_Mg Strength Properties

The shape of the fabricated test specimens for both groups of analysed materials is shown in Figure 2.

The measurement of the strain was carried out by employing a non-contact method using DIC (digital image correlation) technology (Dantec Q450) and confronted with the measurement using an extensometer. The DIC measurement allowed the comparison of the basic material properties: Young’s modulus and Poisson’s ratio. The difference in Poisson ratio (cast 0.35, AM 0.36) was not significant. The Young’s modulus of cast 82 GPa compared to 73 AM already shows a significant difference, which is caused by the absence of eutectically free silicon. The tensile test results of both analysed materials are shown in Figure 3.

The significant increase in yield strength and ultimate strength in favour of AM material was confirmed by the results of other authors [21,22]. This phenomenon is not a general rule for AM materials and occurs rather rarely. The cause is to be sought in the microstructure of the material, which is significantly different from conventional casting compared to additive technology (Figure 1). The grain boundary cohesion and the influence of intermetallic particles on it play an important role in the strength properties, which, on the one hand, increase Young’s modulus but, on the other hand, have an opposite effect on the conventional strength of this material.

### 3.2. Analysis of AlSi_10_Mg Cyclic Properties

Several authors have published the results of cyclic tests on samples of AlSi_10_Mg material produced additively [23,24,25]. An outstanding property of metals that is self-evident and frequently exploited in practice is their isotropy, a useful and crucial metal property whose partial loss may be caused by the additive manufacturing technology of metals. In strength properties, anisotropy is less significant from the point of view of engineering practice. It has only a minimal effect on yield strength, as can be seen in Table 2.

The effect of the direction of cyclic loading versus the direction of deposition of the material layers was measured on two groups of specimens that differed in the direction of printing (Figure 4).

The results of the cyclic tests in the controlled force mode in the form of Basquin relations [27] are shown in Figure 5.

The difference in the cyclic properties caused by the direction of printing of the material of the test specimens was significant (statistically speaking, the difference exceeds the limits of the confidence intervals of the curves). The material built in planes along the horizontal axis of the specimen was cyclically stronger when loaded in the vertical direction (Figure 4) compared to building along the vertical axis. This fact is probably due to the more favourable orientation of defects with respect to their propagation in the microstructure of the material for a given loading direction (Figure 6).

The vast majority of defects are represented by craters after a missing grain of powder. Their geometry (Figure 7b) is relatively favourable in terms of stress concentration. They do not pose a risk of microcrack formation, and when the crack propagates through such a defect shape, it can stop due to the favourable rounding radius at the tip of the crack. Fatigue life is significantly shortened by sharp defects (Figure 7c), especially if they are oriented perpendicular to the direction of the main normal stress.

It is important to compare these cyclic properties in the two perpendicular planes of the AM material build-up with that of the conventionally cast material. All three materials were cyclically loaded in the state without any heat treatment with the same surface treatment of the specimens during their fabrication (grinding).

The course of the curves in Figure 8 indicates significantly lower cycling parameters of the materials produced by AM technology. The Basquin equation parameters of these S-N curves are shown in Table 3 for the regression lines and for the lines representing the prediction interval [29].

## 4. Discussion

The increase in strength properties of AlSi_10_Mg alloy produced additively compared to conventional metallurgy may be related to the lower melting temperature of Al and hence the greater depth of the melting pools in the powder melting process. The high thermal conductivity coefficient of aluminium causes repeated thermal effects on the layers below the deposition layer and the corresponding surroundings, but the high heat transfer rate will not allow the formation of intermetallic Mg_2_Si particles. Thus, the alloy remains in the solid solution phase, and slip processes under cyclic loading do not have additional barriers to movement, which is one of the reasons for the low fatigue properties after additive manufacturing. The repeated thermal influence of the layers with rapid heat dissipation deformationally reinforces the material, which is reflected in its increased strength properties.

Practically all published results confirm the difference in cyclic properties compared to the direction of formation of individual material layers during additive manufacturing [30,31]. This is probably caused by the orientation of sharp defects, which arise mainly at the borders of melting pools. Their orientation is conditioned by the direction of movement of the laser beam, and the effect is multiplied by the observable thickening of the microstructure of the material at these boundaries (Figure 7a).

The direction of loading versus the direction of material deposition during AM in service is difficult to determine. Moreover, it is likely that this direction will not remain constant but will vary. For the computational estimation of the fatigue lifetime, it is, therefore, necessary to assume inferior cyclic properties (vertical deposition direction) for the time being. In the event that with further development of the AM technology, it will be possible to achieve isotropy in the cyclic properties of the AlSi_10_Mg material, the difference between the S-N curve of the AM material (blue line in Figure 8) and the conventionally cast material (black curve in Figure 8) will still remain substantial. The gradual creation of a solid phase from the melt during conventional material production, compared to small melting pools in a solid environment during additive manufacturing, does not create the same resistant microstructure of the material against cyclic loading. In addition, the number of defects creates local stress fields accelerating the formation and growth of microcracks from existing defects.

The different slope of these curves indicates a decreasing difference in the low-cycle region and, conversely, an increasing difference in the high-cycle region. This phenomenon may be due to local plastic deformation at the defect locations at higher cyclic loading levels. This reason is also supported by the fact that AlSi_10_Mg was observed to be a significantly cyclically strengthening material both in the AM condition and after conventional casting [22,31,32].

## 5. Conclusions

From the point of view of the in-service use of components made from AlSi_10_Mg by AM technology, the area of cyclic properties of the material is particularly critical. Taking into consideration experimental measurements performed in this study and the comparison with previous results of other authors in this field, we can formulate the following conclusions:Static tests of AlSi_10_Mg show lower values of Young’s modulus in AM technologies compared to conventional casting of the material, but on the contrary, an increase of strength in AM technologies.Cyclic tests of AMSi_10_Mg show a significant anisotropy depending on the direction of material addition for AM technology and a significant decrease in cyclic properties to the detriment of AM technology.The theoretical achievement of cyclic property isotropy by advances in AM technology does not erase the difference from the cyclic properties of convectively cast material, which is likely due to the physical limits of current AM technologies.

After exhausting all possibilities for optimizing the additive manufacturing parameters, some hope is presented by suitable heat treatment, which can improve the cyclic properties of AlSi_10_Mg [15,33,34,35]. Both AM and conventional casting technologies have this possibility, and finding optimal heat treatment parameters, specifically for AM material, will have to be the subject of further research.

## Figures and Tables

**Figure 1 materials-16-02598-f001:**
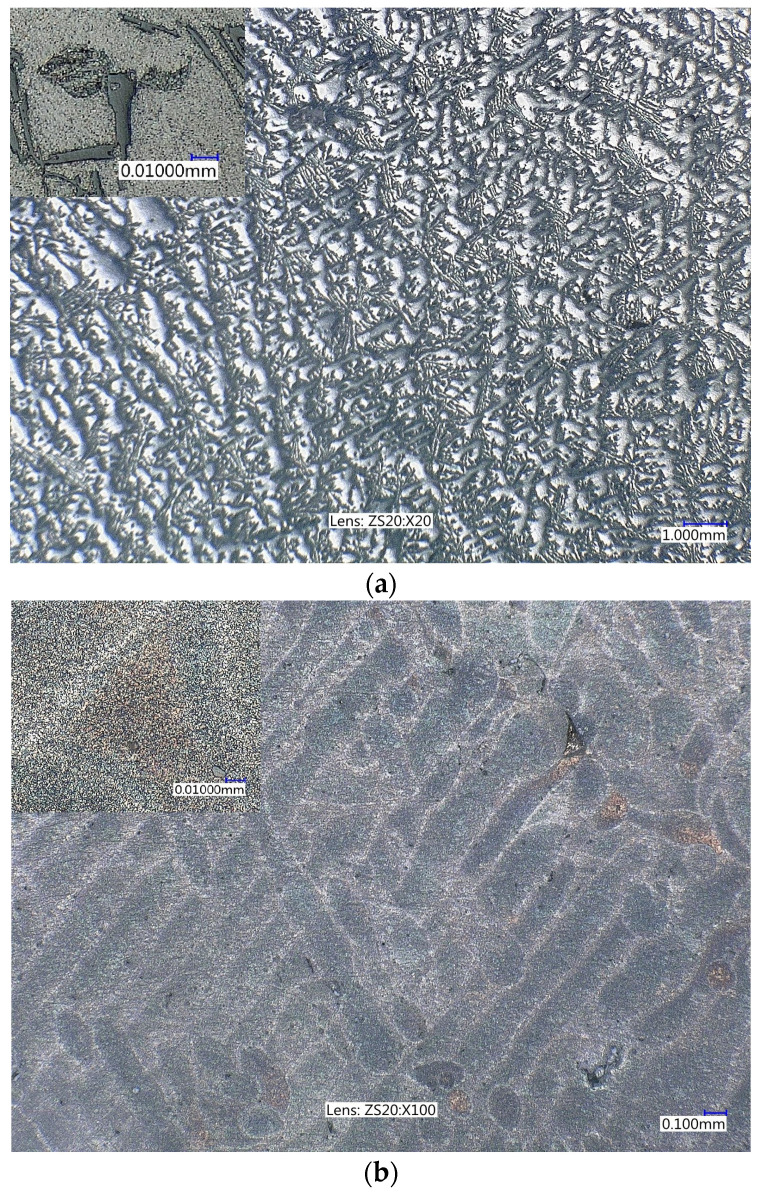
Microstructure of AlSi_10_Mg: (**a**) conventional casting, (**b**) AM technology. Etched in Hypofluorous acid (HF) and 10% sodium hydroxide (NaOH).

**Figure 2 materials-16-02598-f002:**
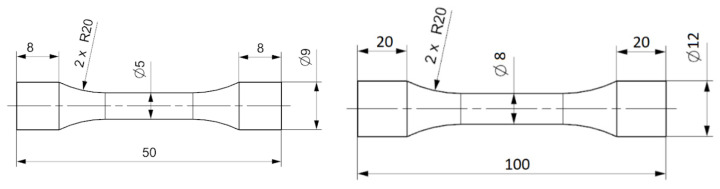
Shape of samples produced additively (**left**) and conventionally (**right**).

**Figure 3 materials-16-02598-f003:**
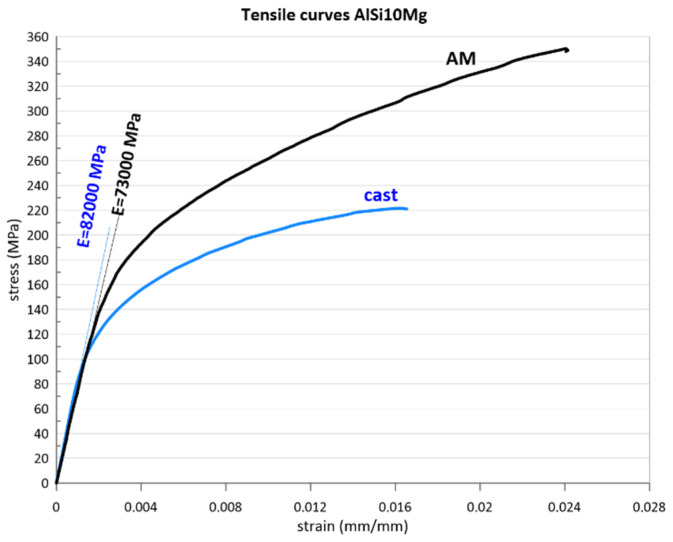
Resulting tensile test diagrams of AlSi_10_Mg material.

**Figure 4 materials-16-02598-f004:**
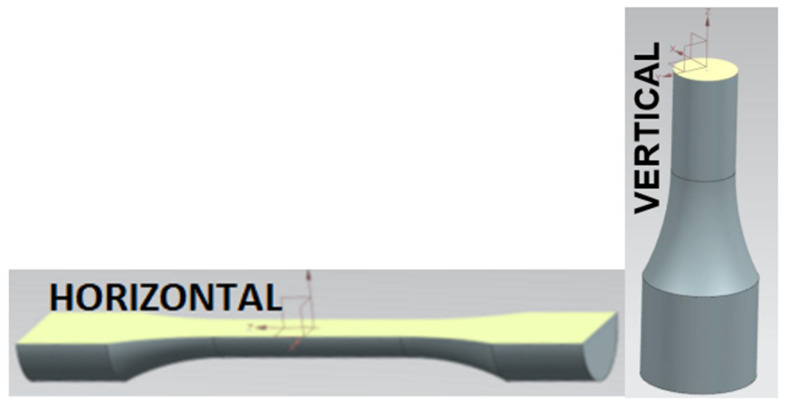
Directions of material layering in the production of the test specimens.

**Figure 5 materials-16-02598-f005:**
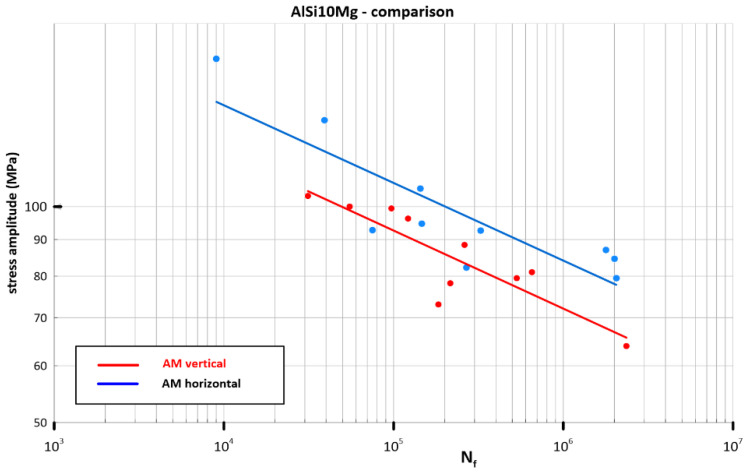
S-N curves of AlSi_10_Mg as a function of the deposition direction.

**Figure 6 materials-16-02598-f006:**
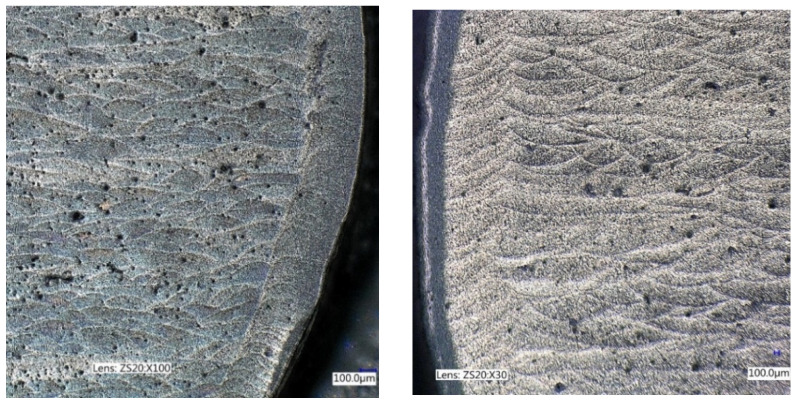
Defects in the specimen cross-section: AM vertical (**left**) and AM horizontal (**right**) [28].

**Figure 7 materials-16-02598-f007:**
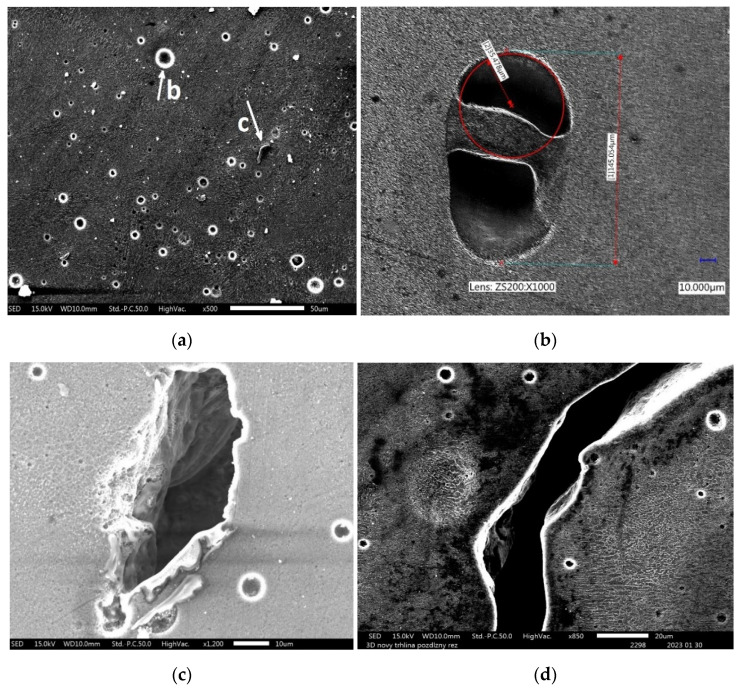
(**a**) Defects in AM material; (**b**) detail of a defect after missing grain of powder; (**c**) detail of a sharp defect; (**d**) part of the fatigue crack trajectory.

**Figure 8 materials-16-02598-f008:**
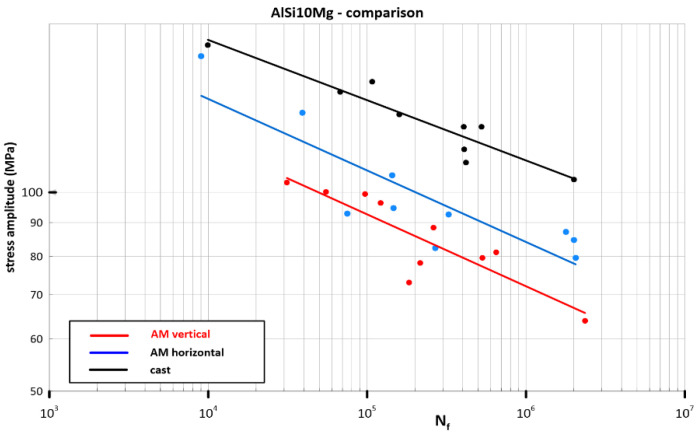
Comparison of AlSi_10_Mg S-N curves of material produced by different technological processes.

**Table 1 materials-16-02598-t001:** Chemical composition of the analysed materials.

	Si (%)	Mg (%)	Fe (%)	Ti (%)	Mn (%)	Cu (%)	Zn (%)	Cr (%)
Conventional metallurgy	10.20	0.346	0.112	0.121	0.046	0.0017	0.02	0.002
Additive manufacturing	10.1	0.38	0.09	<0.03	<0.03	<0.03	<0.03	-

**Table 2 materials-16-02598-t002:** Anisotropy of strength properties AlSi_10_Mg (is in accordance with [26]).

AlSi_10_Mg	45°	90°	180°
Ultimate stress (MPa)	360–370 (367)	325–390 (366)	316–350 (336)
Yield stress (MPa)	230–268 (252)	220–260 (248)	224–250 (236)
Ductility (%)	6.0–9.6 (8.2)	5.6–10.0 (7.63)	10.0–16.3 (12.83)

**Table 3 materials-16-02598-t003:** Cyclic material parameters.

Material	Cyclic Axial					
	RL		97.5% PI		2.5% PI	
	σ_f’_ (MPa)	b_σ_ (-)	σ_f’_ (MPa)	b_σ_ (-)	σ_f´_ (MPa)	b_σ_ (-)
AM_Vertical	553	−0.1442	595	−0.1404	511	−0.1483
AM_Horizontal	651	−0.1449	956	−0.1587	472	−0.1332
Cast	488	−0.1028	539	−0.1044	443	−0.1012

## Data Availability

Not applicable.

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
