# Peer review of "Strength and Cyclic Properties of Additive vs. Conventionally Produced Material AlSi10Mg"

_materials, 2023, doi:10.3390/ma16072598_

Round 1
Reviewer 1 Report
This article experimentally analyses strength properties and fatigue properties in the tensile-pressure mode for two different directions of 3D printing of AlSi10Mg material. The resulting fatigue parameters of the Basquin curve are confronted with a conventionally produced alloy of the same composition. The main comments are as follows:
1. When the mechanical properties of additive and traditional AlSi10Mg alloys are tested and compared, why are the sample sizes used different (Fig.2)? The different test sample sizes will affect the performance comparison. How to consider this problem?
2. There is no scale in Fig. 6. It can be seen from Fig. 6 that there are a lot of defects in the additive alloy materials, and the authors think that they will not cause the risk of microcrack formation. I think this conclusion is incorrect, and the defects must have an impact on the properties of the materials. If the author believes that the defect does not pose a risk of microcrack formation, the author is requested to provide evidence.
3. The ruler in Figure 7 is inconsistent and not obvious. It is recommended to re-label.
4. In this study, the properties of materials obtained by additive manufacturing are lower than those obtained by as cast. However, there are a large number of defects in the alloy materials prepared by additive manufacturing in this paper, which will affect the properties of the materials. It is not appropriate to compare the properties of a defective material with those of other materials, so the conclusions drawn are not general. I think it is very important to improve the process of additive alloy manufacturing and reduce defects in the future.
5. To sum up, I think this paper is not innovative, the conclusion has problems, do not recommend accepted.
Author Response
Dear reviewer, thank you for your time and I appreciate the comments on our article. The responses are in the attached file.

Reviewer 2 Report
The article is interesting and contains many interesting topics. This is an article that would be suitable for publication but should be improved.
List of issues requiring improvement:
1. The introduction is quite short for an article in such a reputable journal as Materials.
2. Line 48 has the abbreviation "SLM" and line 83 has the abbreviation "DIC" which may be incomprehensible to some readers. Please complete the full name.
3. Are Figure 1 made by the authors of the publication. There is no information on how to do it. If these are Figures from other publications, this should be a reference.
4. Please explain "The significant increase in yield strength and ultimate strength in favor of AM material was confirmed by the results of other authors [15, 16]. This phenomenon is not a general rule for AM materials and occurs rather rarely." Do many studies confirm this, and the phenomenon is rare?
5. "There is some hope for appropriate heat treatment that can enhance the cyclic 194 properties of AlSi10Mg [27-30]." This sentence is not the result of the research presented in the article. This statement is based on research by other authors and should not be included in the conclusions of this paper.
Author Response

(The authors gave the same response as above.)

Reviewer 3 Report
I have read the article entitled “Strength and cyclic properties of additive vs. conventionally produced material AlSi10Mg” is an interesting and is having some publishable information. I have observed the following points. The authors have to address on it before it is being accepted.
1. Abstract: The present form of abstract is like general one. Here, the authors are asked to mention their contribution/specific parameters and outcomes with scientific reasons. It has to be revised.
2. Introduction part is very weak. Some more literature discussion are to be incorporated.
3. The main objectives of the present work is to be incorporated at the end of introduction part which is missing in present form of manuscript.
4. A schematic diagram representing the present work is to be incorporated in Methodology section. This will enrich the quality of the paper.
5. Phases/features observed in Fig.1 are to marked in the images
6. Based on the schematic of Tensile test samples, why the authors have taken different geometry? Will the geometry not affect the mechanical behaviour? Need clarification
7. Various defects observed from Figs 6 & 7 are to be marked in the corresponding images.
Author Response

(The authors gave the same response as above.)

Author Response

(The authors gave the same response as above.)

Round 2
Reviewer 3 Report
The authors have revised the article as per my comments and hence, I am recommending to accept the revised version.
Reviewer 4 Report
Dear authors, I have read your revised manuscript and I found it more sound than the previous version. You answered clearly and satisfactory to all my previous review remarks. I recommend publishing the paper in the MDPI journal as it is.